# Revisiting Bayes by Backprop

## Abstract

In this work we explore a straightforward variational Bayes scheme for Recurrent Neural Networks. Firstly, we show that a simple adaptation of truncated backpropagation through time can yield good quality uncertainty estimates and superior regularisation at only a small extra computational cost during training, also reducing the amount of parameters by 80%. Secondly, we demonstrate how a novel kind of posterior approximation yields further improvements to the performance of Bayesian RNNs. We incorporate local gradient information into the approximate posterior to sharpen it around the current batch statistics. We show how this technique is not exclusive to recurrent neural networks and can be applied more widely to train Bayesian neural networks. We also empirically demonstrate how Bayesian RNNs are superior to traditional RNNs on a language modelling benchmark and an image captioning task, as well as showing how each of these methods improve our model over a variety of other schemes for training them. We also introduce a new benchmark for studying uncertainty for language models so future methods can be easily compared.

## 1 Introduction

Recurrent Neural Networks (RNNs) achieve state-of-the-art performance on a wide range of sequence prediction tasks (Wu et al., 2016; Amodei et al., 2015; Jozefowicz et al., 2016; Zaremba et al., 2014; Lu et al., 2016). In this work we examine how to add uncertainty and regularisation to RNNs by means of applying Bayesian methods to training. This approach allows the network to express uncertainty via its parameters. At the same time, by using a prior to integrate out the parameters to average across many models during training, it gives a regularisation effect to the network. Recent approaches either justify dropout (Srivastava et al., 2014) and weight decay as a variational inference scheme (Gal & Ghahramani, 2016), or apply Stochastic Gradient Langevin dynamics (Welling & Teh, 2011, SGLD) to truncated backpropagation in time directly (Gan et al., 2016). Interestingly, recent work has not explored further directly applying a variational Bayes inference scheme (Beal, 2003) for RNNs as was done in Graves (2011). We derive a straightforward approach based upon Bayes by Backprop (Blundell et al., 2015) that we show works well on large scale problems. Our strategy is a simple alteration to truncated backpropagation through time that results in an estimate of the posterior distribution on the weights of the RNN. This formulation explicitly leads to a cost function with an information theoretic justification by means of a bits-back argument (Hinton & Van Camp, 1993) where a KL divergence acts as a regulariser.

The form of the posterior in variational inference shapes the quality of the uncertainty estimates and hence the overall performance of the model. We shall show how performance of the RNN can be improved by means of adapting ("sharpening") the posterior locally to a batch. This sharpening adapts the variational posterior to a batch of data using gradients based upon the batch. This can be viewed as a hierarchical distribution, where a local batch gradient is used to adapt a global posterior, forming a local approximation for each batch. This gives a more flexible form to the typical assumption of Gaussian posterior when variational inference is applied to neural networks, which reduces variance. This technique can be applied more widely across other Bayesian models.

The contributions of our work are as follows:

- We show how Bayes by Backprop (BBB) can be efficiently applied to RNNs.
- We develop a novel technique which reduces the variance of BBB, and which can be widely adopted in other maximum likelihood frameworks.

- We improve performance on two widely studied benchmarks outperforming established regularisation techniques such as dropout by a big margin.
- We introduce a new benchmark for studying uncertainty of language models.

## 2  BAYES BY BACKPROP

Bayes by Backprop (Graves, 2011; Blundell et al., 2015) is a variational inference (Wainwright et al., 2008) scheme for learning the posterior distribution on the weights $\theta \in \mathbb{R}^d$ of a neural network. This posterior distribution is typically taken to be a Gaussian with mean parameter $\mu \in \mathbb{R}^d$ and standard deviation parameter $\sigma \in \mathbb{R}^d$, denoted $\mathcal{N}(\theta|\mu, \sigma^2)$. Note that we use a diagonal covariance matrix, and $d$ – the dimensionality of the parameters of the network – is typically in the order of millions. Let $\log p(y|\theta, x)$ be the log-likelihood of the model, then the network is trained by minimising the variational free energy:

$$\mathcal{L}(\theta) = \mathbb{E}_{q(\theta)}\left[\log \frac{q(\theta)}{p(y|\theta, x)p(\theta)}\right], \tag{1}$$

where $p(\theta)$ is a prior on the parameters.

Minimising the variational free energy (1) is equivalent to maximising the log-likelihood $\log p(y|\theta, x)$ subject to a KL complexity term on the parameters of the network that acts as a regulariser:

$$\mathcal{L}(\theta) = -\mathbb{E}_{q(\theta)}\left[\log p(y|\theta, x)\right] + \text{KL}\left[q(\theta) \,||\, p(\theta)\right]. \tag{2}$$

In the Gaussian case with a zero mean prior, the KL term can be seen as a form of weight decay on the mean parameters, where the rate of weight decay is automatically tuned by the standard deviation parameters of the prior and posterior. Please refer to the supplementary material for the algorithmic details on Bayes by Backprop.

The uncertainty afforded by Bayes by Backprop trained networks has been used successfully for training feedforward models for supervised learning and to aid exploration by reinforcement learning agents (Blundell et al., 2015; Lipton et al., 2016; Houthooft et al., 2016), but as yet, it has not been applied to recurrent neural networks.

## 3  TRUNCATED BAYES BY BACKPROP THROUGH TIME

The core of an RNN, $f$, is a neural network that maps the RNN state $s_t$ at step $t$, and an input observation $x_t$ to a new RNN state $s_{t+1}$, $f : (s_t, x_t) \mapsto s_{t+1}$. The exact equations of an LSTM core can be found in the supplemental material Sec A.2.

An RNN can be trained on a sequence of length $T$ by backpropagation through by unrolling $T$ times into a feedforward network. Explicitly, we set $s_i = f(s_{i-1}, x_i)$, for $i = 1, \ldots, T$. We shall refer to an RNN core unrolled for $T$ steps by $s_{1:T} = F_T(x_{1:T}, s_0)$. Note that the truncated version of the algorithm can be seen as taking $s_0$ as the last state of the previous batch, $s_T$.

RNN parameters are learnt in much the same way as in a feedforward neural network. A loss (typically after further layers) is applied to the states $s_{1:T}$ of the RNN, and then backpropagation is used to update the weights of the network. Crucially, the weights at each of the unrolled steps are shared. Thus each weight of the RNN core receives $T$ gradient contributions when the RNN is unrolled for $T$ steps.

Applying BBB to RNNs is depicted in Figure 1 where the weight matrices of the RNN are drawn from a distribution (learnt by BBB). However, this direct application raises two questions: when to sample the parameters of the RNN, and how to weight the contribution of the KL regulariser of (2). We shall briefly justify the adaptation of BBB to RNNs, given in Figure 1. The variational free energy of (2) for an RNN on a sequence of length $T$ is:

$$\mathcal{L}(\theta) = -\mathbb{E}_{q(\theta)}\left[\log p(y_{1:T}|\theta, x_{1:T})\right] + \text{KL}\left[q(\theta) \,||\, p(\theta)\right], \tag{3}$$

where $p(y_{1:T}|\theta, x_{1:T})$ is the likelihood of a sequence produced when the states of an unrolled RNN $F_T$ are fed into an appropriate probability distribution. The parameters of the entire network are

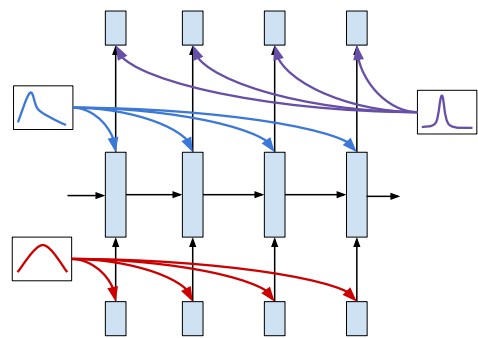

**Algorithm: Bayes by Backprop for RNNs**

Sample $\epsilon \sim \mathcal{N}(0, I)$, $\epsilon \in \mathbb{R}^d$, and set network parameters to $\theta = \mu + \sigma\epsilon$.
Sample a minibatch of truncated sequences $(x, y)$.
Do forward and backward propagation as normal, and let $g$ be the gradient w.r.t $\theta$.
Let $g_\theta^{KL}, g_\mu^{KL}, g_\sigma^{KL}$ be the gradients of $\log\mathcal{N}(\theta|\mu, \sigma^2) - \log p(\theta)$ w.r.t. $\theta$, $\mu$ and $\sigma$ respectively.
Update $\mu$ using the gradient $\frac{g + \frac{1}{C}g_\theta^{KL}}{B} + \frac{g_\mu^{KL}}{BC}$.
Update $\sigma$ using the gradient $\left(\frac{g + \frac{1}{C}g_\theta^{KL}}{B}\right)\epsilon + \frac{g_\sigma^{KL}}{BC}$.

Figure 1: Illustration (left) and Algorithm (right) of Bayes by Backprop applied to an RNN.

$\theta$. Although the RNN is unrolled $T$ times, each weight is penalised just once by the KL term, rather than $T$ times. Also clear from (3) is that when a Monte Carlo approximation is taken to the expectation, the parameters $\theta$ should be held fixed throughout the entire sequence.

Two complications arise to the above naive derivation in practice: firstly, sequences are often long enough and models sufficiently large, that unrolling the RNN for the whole sequence is prohibitive. Secondly, to reduce variance in the gradients, more than one sequence is trained at a time. Thus the typical regime for training RNNs involves training on mini-batches of truncated sequences.

Let $B$ be the number of mini-batches and $C$ the number of truncated sequences ("cuts"), then we can write (3) as:

$$\mathcal{L}(\theta) = -\mathbb{E}_{q(\theta)}\left[\log\prod_{b=1}^{B}\prod_{c=1}^{C}p(y^{(b,c)}|\theta, x^{(b,c)})\right] + \mathrm{KL}\left[q(\theta) \,||\, p(\theta)\right], \tag{4}$$

where the $(b, c)$ superscript denotes elements of $c$th truncated sequence in the $b$th minibatch. Thus the free energy of mini-batch $b$ of a truncated sequence $c$ can be written as:

$$\mathcal{L}_{(b,c)}(\theta) = -\mathbb{E}_{q(\theta)}\left[\log p(y^{(b,c)}|\theta, x^{(b,c)}, s_{\mathrm{prev}}^{(b,c)})\right] + w_{\mathrm{KL}}^{(b,c)}\mathrm{KL}\left[q(\theta) \,||\, p(\theta)\right], \tag{5}$$

where $w_{\mathrm{KL}}^{(b,c)}$ distributes the responsibility of the KL cost among minibatches and truncated sequences (thus $\sum_{b=1}^{B}\sum_{c=1}^{C} w_{\mathrm{KL}}^{(b,c)} = 1$), and $s_{\mathrm{prev}}^{(b,c)}$ refers to the initial state of the RNN for the minibatch $x^{(b,c)}$. In practice, we pick $w_{\mathrm{KL}}^{(b,c)} = \frac{1}{CB}$ so that the KL penalty is equally distributed among all mini-batches and truncated sequences. The truncated sequences in each subsequent mini-batches are picked in order, and so $s_{\mathrm{prev}}^{(b,c)}$ is set to the last state of the RNN for $x^{(b,c-1)}$.

Finally, the question of when to sample weights follows naturally from taking a Monte Carlo approximations to (5): for each minibatch, sample a fresh set of parameters.

## 4 POSTERIOR SHARPENING

The choice of variational posterior $q(\theta)$ as described in Section 3 can be enhanced by adding side information that makes the posterior over the parameters more accurate, thus reducing variance of the learning process.

Akin to Variational Auto Encoders (VAEs) (Kingma & Welling, 2013; Rezende et al., 2014), which propose a powerful distribution $q(z|x)$ to improve the gradient estimates of the (intractable) likelihood function $p(x)$, here we propose a similar approach. Namely, for a given minibatch of data (inputs and targets) $(x, y)$ sampled from the training set, we construct such $q(\theta|(x, y))$. Thus, we compute a proposal distribution where the latents ($z$ in VAEs) are the parameters $\theta$ (which we wish to integrate out), and the "privileged" information upon which we condition is a minibatch of data.

We could have chosen to condition on a single example $(x, y)$ instead of a batch, but this would have yielded different parameter vectors $\theta$ per example. Conditioning on the full minibatch has

the advantage of producing a single $\theta$ per minibatch, so that matrix-matrix operations can still be carried.

This "sharpened" posterior yields more stable optimisation, a common pitfall of Bayesian approaches to train neural networks, and the justification of this method follows from strong empirical evidence and extensive work on VAEs.

A challenging aspect of modelling the variational posterior $q(\theta|(x, y))$ is the large number of dimensions of $\theta \in \mathbb{R}^d$. When the dimensionality is not in the order of millions, a powerful non-linear function (such as a neural network) can be used which transforms observations $(x, y)$ to the parameters of a Gaussian distribution, as proposed in Kingma & Welling (2013); Rezende et al. (2014). Unfortunately, this neural network would have far too many parameters, making this approach unfeasible.

Given that the loss $-\log p(y|\theta, x)$ is differentiable with respect to $\theta$, we propose to parameterise $q$ as a linear combination of $\theta$ and $g_\theta = -\nabla_\theta \log p(y|\theta, x)$, both $d$-dimensional vectors.

Thus, we can define a hierarchical posterior of the form

$$q(\theta|(x, y)) = \int q(\theta|\varphi, (x, y))q(\varphi)d\varphi, \tag{6}$$

with $\mu, \sigma \in \mathbb{R}^d$, and $q(\varphi) = \mathcal{N}(\varphi|\mu, \sigma)$ – the same as in the standard BBB method. Finally, let $*$ denote element-wise multiplication, we then have

$$q(\theta|\varphi, (x, y)) = \mathcal{N}(\theta|\varphi - \eta * g_\varphi, \sigma_0^2 I), \tag{7}$$

where $\eta \in \mathbb{R}^d$ is a free parameter to be learnt and $\sigma_0$ a scalar hyper-parameter of our model. $\eta$ can be interpreted as a per-parameter learning rate.

During training, we get $\theta \sim q(\theta|(x, y))$ via ancestral sampling to optimise the loss

$$L(\mu, \sigma, \eta) = E_{(x,y)}[E_{q(\varphi)q(\theta|\varphi,(x,y))}[L(x, y, \theta, \varphi|\mu, \sigma, \eta)]], \tag{8}$$

with $L(x, y, \theta, \varphi|\mu, \sigma, \eta)$ given by

$$L(x, y, \theta, \varphi|\mu, \sigma, \eta) = -\log p(y|\theta, x) + \text{KL}\left[q(\theta|\varphi, (x, y)) \,||\, p(\theta|\varphi)\right] + \frac{1}{C}\text{KL}\left[q(\varphi) \,||\, p(\varphi)\right], \tag{9}$$

where $\mu, \sigma, \eta$ are our model parameters, and $p$ are the priors for the distributions defining $q$ (for exact details of these distributions see Section 6). The constant $C$ is the number of truncated sequences as defined in Section 3. The bound on the true data likelihood which yields eq. (8) is derived in Sec 4.1. Algorithm 1 presents how learning is performed in practice.

---

**Algorithm 1** BBB with Posterior Sharpening

---

Sample a minibatch $(x, y)$ of truncated sequences.
Sample $\varphi \sim q(\varphi) = \mathcal{N}(\varphi|\mu, \sigma)$.
Let $g_\varphi = -\nabla_\varphi \log p(y|\varphi, x)$.
Sample $\theta \sim q(\theta|\varphi, (x, y)) = \mathcal{N}(\theta|\varphi - \eta * g_\varphi, \sigma_0^2 I)$.
Compute the gradients of eq. (8) w.r.t. $(\mu, \sigma, \eta)$.
Update $(\mu, \sigma, \eta)$.

---

As long as the improvement of the log likelihood $\log p(y|\theta, x)$ term along the gradient $g_\varphi$ is greater than the KL cost added for posterior sharpening (KL $\left[q(\theta|\varphi, (x, y)) \,||\, p(\theta|\varphi)\right]$), then the lower bound in (8) will improve. This justifies the effectiveness of the posterior over the parameters proposed in eq. 7 which will be effective as long as the curvature of $\log p(y|\theta, x)$ is large. Since $\eta$ is learnt, it controls the tradeoff between curvature improvement and KL loss. Studying more powerful parameterisations is part of future research.

Unlike regular BBB where the KL terms can be ignored during inference, there are two options for doing inference under posterior sharpening. The first involves using $q(\varphi)$ and ignoring any KL terms, similar to regular BBB. The second involves using $q(\theta|\varphi, (x, y))$ which requires using the term KL $\left[q(\theta|\varphi, (x, y)) \,||\, p(\theta|\varphi)\right]$ yielding an upper bound on perplexity (lower bound in log probability; see Section 4.2 for details). This parameterisation involves computing an extra gradient and incurs a penalty in training speed. A comparison of the two inference methods is provided in Section 6. Furthermore, in the case of RNNs, the exact gradient cannot be efficiently computed, so BPTT is used.

### 4.1 DERIVATION OF FREE ENERGY FOR POSTERIOR SHARPENING

Here we turn to deriving the training loss function we use for posterior sharpening. The basic idea is to take a variational approximation to the marginal likelihood $p(x)$ that factorises hierarchically. Hierarchical variational schemes for topic models have been studied previously in Ranganath et al. (2016). Here, we shall assume a hierarchical prior for the parameters such that $p(x) = \int p(x|\theta)p(\theta|\varphi)p(\varphi)d\theta d\varphi$. Then we pick a variational posterior that conditions upon $x$, and factorises as $q(\theta, \varphi|x) = q(\theta|\varphi, x)q(\varphi)$. The expected lower bound on $p(x)$ is then as follows:

$$\log p(x) = \log \left( \int p(x|\theta)p(\theta|\varphi)p(\varphi)d\theta d\varphi \right) \tag{10}$$

$$\geq \mathbb{E}_{q(\varphi, \theta|x)} \left[ \log \frac{p(x|\theta)p(\theta|\varphi)p(\varphi)}{q(\varphi, \theta|x)} \right] \tag{11}$$

$$= \mathbb{E}_{q(\theta|\varphi, x)q(\varphi)} \left[ \log \frac{p(x|\theta)p(\theta|\varphi)p(\varphi)}{q(\theta|\varphi, x)q(\varphi)} \right] \tag{12}$$

$$= \mathbb{E}_{q(\varphi)} \left[ \mathbb{E}_{q(\theta|\varphi, x)} \left[ \log p(x|\theta) + \log \frac{p(\theta|\varphi)}{q(\theta|\varphi, x)} \right] + \log \frac{p(\varphi)}{q(\varphi)} \right] \tag{13}$$

$$= \mathbb{E}_{q(\varphi)} \left[ \mathbb{E}_{q(\theta|\varphi, x)} [\log p(x|\theta)] - \mathrm{KL}\left[ q(\theta|\varphi, x) \,||\, p(\theta|\varphi) \right] \right] - \mathrm{KL}\left[ q(\varphi) \,||\, p(\varphi) \right] \tag{14}$$

### 4.2 DERIVATION OF PREDICTIONS WITH POSTERIOR SHARPENING

Now we consider making predictions. These are done by means of Bayesian model averaging over the approximate posterior. In the case of no posterior sharpening, predictions are made by evaluating: $\mathbb{E}_{q(\theta)} [\log p(\hat{x}|\theta)]$. For posterior sharpening, we derive a bound on a Bayesian model average over the approximate posterior of $\varphi$:

$$\mathbb{E}_{q(\varphi)} [\log p(\hat{x}|\varphi)] = \mathbb{E}_{q(\varphi)} \left[ \log \int p(\hat{x}|\theta)p(\theta|\varphi)d\theta \right] \tag{15}$$

$$\geq \mathbb{E}_{q(\varphi)} \left[ \mathbb{E}_{q(\theta|\varphi, x)} \left[ \log \frac{p(\hat{x}|\theta)p(\theta|\varphi)}{q(\theta|\varphi, x)} \right] \right] \tag{16}$$

$$= \mathbb{E}_{q(\varphi)} \left[ \mathbb{E}_{q(\theta|\varphi, x)} [\log p(\hat{x}|\theta)] - \mathrm{KL}\left[ q(\theta|\varphi, x) \,||\, p(\theta|\varphi) \right] \right] \tag{17}$$

## 5 RELATED WORK

We note that the procedure of sharpening the posterior as explained above has similarities with other techniques. Perhaps the most obvious one is line search: indeed, $\eta$ is a trained parameter that does line search along the gradient direction. Probabilistic interpretations have been given to line search in e.g. Mahsereci & Hennig (2015), but ours is the first that uses a variational posterior with the reparametrization trick/perturbation analysis gradient. Also, the probabilistic treatment to line search can also be interpreted as a trust region method.

Another related technique is dynamic evaluation (Mikolov et al., 2010), which trains an RNN during evaluation of the model with a fixed learning rate. The update applied in this case is cumulative, and only uses previously seen data. Thus, they can take a purely deterministic approach and ignore any KL between a posterior with privileged information and a prior. As we will show in Section 6, performance gains can be significant as the data exhibits many short term correlations.

Lastly, learning to optimise (or learning to learn) (Li & Malik, 2016; Andrychowicz et al., 2016) is related in that a learning rate is learned so that it produces better updates than those provided by e.g. AdaGrad (Duchi et al., 2011) or Adam (Kingma & Ba, 2014). Whilst they train a parametric model, we treat these as free parameters (so that they can adapt more quickly to the non-stationary distribution w.r.t. parameters). Notably, we use gradient information to inform a variational posterior so as to reduce variance of Bayesian Neural Networks. Thus, although similar in flavour, the underlying motivations are quite different.

Applying Bayesian methods to neural networks has a long history, with most common approximations having been tried. Buntine & Weigend (1991) propose various maximum a posteriori schemes

for neural networks, including an approximate posterior centered at the mode. Buntine & Weigend (1991) also suggest using second order derivatives in the prior to encourage smoothness of the resulting network. Hinton & Van Camp (1993) proposed using variational methods for compressing the weights of neural networks as a regulariser. Hochreiter et al. (1995) suggest an MDL loss for single layer networks that penalises non-robust weights by means of an approximate penalty based upon perturbations of the weights on the outputs. Denker & Lecun (1991); MacKay (1995) investigated using the Laplace approximation for capturing the posterior of neural networks. Neal (2012) investigated the use of hybrid Monte Carlo for training neural networks, although it has so far been difficult to apply these to the large sizes of networks considered here.

More recently Graves (2011) derived a variational inference scheme for neural networks and Blundell et al. (2015) extended this with an update for the variance that is unbiased and simpler to compute. Graves (2016) derives a similar algorithm in the case of a mixture posterior. Several authors have claimed that dropout (Srivastava et al., 2014) and Gaussian dropout (Wang & Manning, 2013) can be viewed as approximate variational inference schemes (Gal & Ghahramani, 2015; Kingma et al., 2015, respectively).

A few papers have investigated approximate Bayesian recurrent neural networks. Mirikitani & Nikolaev (2010) proposed a second-order, online training scheme for recurrent neural networks, while Chien & Ku (2016) only capture a single point estimate of the weight distribution. Gal & Ghahramani (2016) highlighted Monte Carlo dropout for LSTMs (we explicitly compare to these results in our experiments), whilst Graves (2011) proposed a variational scheme with biased gradients for the variance parameter using the Fisher matrix. Our work extends this by using an unbiased gradient estimator without need for approximating the Fisher and also add a novel posterior approximation. Variational methods typically underestimate the uncertainty in the posterior (as they are mode seeking, akin to the Laplace approximation), whereas expectation propagation methods often average over modes and so tend to overestimate uncertainty (although there are counter examples for each depending upon the particular factorisation and approximations used; see for example (Turner & Sahani, 2011)). Nonetheless, several papers explore applying expectation propagation to neural networks: Soudry et al. (2014) derive a closed form approximate online expectation propagation algorithm, whereas Hernández-Lobato & Adams (2015) proposed using multiple passes of assumed density filtering (in combination with early stopping) attaining good performance on a number of small data sets. Hasenclever et al. (2015) derive a distributed expectation propagation scheme with SGLD (Welling & Teh, 2011) as an inner loop. Others have also considered applying SGLD to neural networks (Li et al., 2015) and Gan et al. (2016) more recently used SGLD for LSTMs (we compare to these results in our experiments).

## 6 EXPERIMENTS

We present the results of our method for a language modelling and an image caption generation task.

### 6.1 LANGUAGE MODELLING

We evaluated our model on the Penn Treebank Marcus et al. (1993) benchmark, a task consisting on next word prediction. We used the network architecture from Zaremba et al. (2014), a simple yet strong baseline on this task, and for which there is an open source implementation[1]. The baseline consists of an RNN with LSTM cells and a special regularisation technique, where the dropout operator is only applied to the non-recurrent connections. We keep the network configuration unchanged, but instead of using dropout we apply our Bayes by Backprop formulation. Our goal is to demonstrate the effect of applying BBB to a pre-existing, well studied architecture.

To train our models, we tuned the parameters on the prior distribution, the learning rate and its decay. The weights were initialised randomly and we used gradient descent with gradient clipping for optimisation, closely following Zaremba et al. (2014)'s "medium" LSTM configuration (2 layers with 650 units each).

---

[1]`https://github.com/tensorflow/models/blob/master/tutorials/rnn/ptb/ptb_word_lm.py`

Table 1: Word-level perplexity on the Penn Treebank language modelling task (lower is better), where DE indicates that Dynamic Evaluation was used.

| Model (medium) | Val | Test | Val (DE) | Test (DE) |
|---|---|---|---|---|
| LSTM (Zaremba et al., 2014) | 120.7 | 114.5 | - | - |
| LSTM dropout (Zaremba et al., 2014) | 86.2 | 82.1 | 79.7 | 77.1 |
| Variational LSTM (tied weights) (Gal & Ghahramani, 2016) | 81.8 | 79.7 | - | - |
| Variational LSTM (tied weights, MS) (Gal & Ghahramani, 2016) | - | 79.0 | - | - |
| Bayesian RNN (BRNN) | 78.8 | 75.5 | 73.4 | 70.7 |
| BRNN w/ Posterior Sharpening | $\leq \mathbf{77.8}$ | $\leq \mathbf{74.8}$ | $\leq \mathbf{72.6}$ | $\leq \mathbf{69.8}$ |

As in Blundell et al. (2015), the prior of the network weights $\theta$ was taken to be a scalar mixture of two Gaussian densities with zero mean and variances $\sigma_1^2$ and $\sigma_2^2$, explicitly

$$p(\theta) = \prod_j \left( \pi \mathcal{N}(\theta_j | 0, \sigma_1^2) + (1 - \pi) \mathcal{N}(\theta_j | 0, \sigma_2^2) \right), \qquad (18)$$

where $\theta_j$ is the $j$-th weight of the network. We searched $\pi \in \{0.25, 0.5, 0.75\}$, $\log \sigma_1 \in \{0, -1, -2\}$ and $\log \sigma_2 \in \{-6, -7, -8\}$.

For speed purposes, during training we used one sample from the posterior for estimating the gradients and computing the (approximate) KL-divergence. For prediction, we experimented with either computing the expected loss via Monte Carlo sampling, or using the mean of the posterior distribution as the parameters of the network (MAP estimate). We observed that the results improved as we increased the number of samples but they were not significantly better than taking the mean (as was also reported by Graves (2011); Blundell et al. (2015)). For convenience, in Table 1 we report our numbers using the mean of the converged distribution, as there is no computation overhead w.r.t. a standard LSTM model.

Table 1 compares our results to the LSTM dropout baseline Zaremba et al. (2014) we built from, and to the Variational LSTMs Gal & Ghahramani (2016), which is another Bayesian approach to this task. Finally, we added dynamic evaluation Mikolov et al. (2010) results with a learning rate of 0.1, which was found via cross validation.

As with other VAE-related RNNs Fabius & van Amersfoort (2014); Bayer & Osendorfer (2014); Chung et al. (2015) perplexities using posterior sharpening are reported including a KL penalty $\mathrm{KL}\left[q(\theta|\varphi, (x, y)) \,\|\, p(\theta|\varphi)\right]$ in the log likelihood term (the KL is computed exactly, not sampled). For posterior sharpening we use a hierarchical prior for $\theta$: $p(\theta|\varphi) = \mathcal{N}(\theta|\varphi, \sigma_0^2 I)$ which expresses our belief that a priori, the network parameters $\theta$ will be much like the data independent parameters $\varphi$ with some small Gaussian perturbation. In our experiments we swept over $\sigma_0$ on the validation set, and found $\sigma_0 = 0.02$ to perform well, although results were not particularly sensitive to this. Note that with posterior sharpening, the perplexities reported are upper bounds (as the likelihoods are lower bounds).

Lastly, we tested the variance reduction capabilities of posterior sharpening by analysing the perplexity attained by the best models reported in Table 1. Standard BBB yields 258 perplexity after only one epoch, whereas the model with posterior sharpening is better at 227. We also implemented it on MNIST following Blundell et al. (2015), and obtained small but consistent speed ups. Lower perplexities on the Penn Treebank task can be achieved by varying the model architecture, which should be complementary to our work of treating weights as random variables—we are simply interested in assessing the impact of our method on an existing architecture, rather than absolute state-of-the-art. See Kim et al. (2015); Zilly et al. (2016); Merity et al. (2016), for a report on recent advances on this benchmark, where they achieve perplexities of 70.9 on the test set. Furthermore we note that the speed of our naïve implementation of Bayesian RNNs was 0.7 times the original speed and 0.4 times the original speed for posterior sharpening. Notably, Figure 2 shows the effect of weight pruning: weights were ordered by their signal-to-noise ratio ($|\mu_i|/\sigma_i$) and removed (set

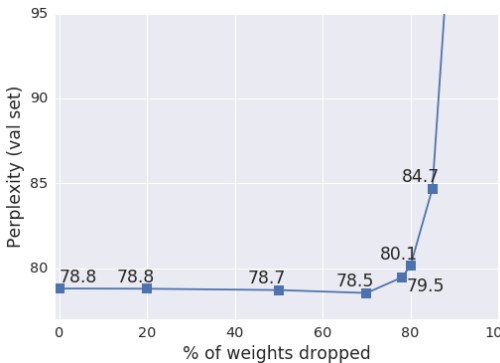 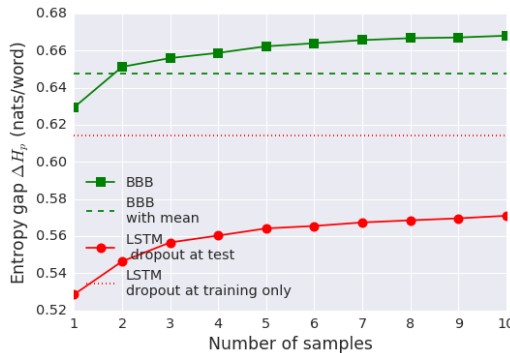

Figure 2: Weight pruning experiment. No significant loss on performance is observed until pruning more than 80% of weights.

Figure 3: Entropy gap $\Delta H_p$ (Eq. (20)) between reversed and regular Penn Treebank test sets $\times$ number of samples.

to zero) in reverse order. We evaluated the validation set perplexity for each proportion of weights dropped. As can be seen, around 80% of the weights can be removed from the network with little impact on validation perplexity. Additional analysis on the existing patterns of the dropped weights can be found in the supplementary material A.3.

### 6.1.1 UNCERTAINTY ANALYSIS

We used the Penn Treebank test set, which is a long sequence of $\approx$ 80K words, and reversed it. Thus, the "reversed" test set first few words are: "us with here them see not may we ..." which correspond to the last words of the standard test set: "... we may not see them here with us".

Let $V$ be the vocabulary of this task. For a given input sequence $x = x_{1:T}$ and a probabilistic model $p$, we define the entropy of $x$ under $p$, $H_p[x]$, by

$$H_p[x] = - \sum_{i=1,...,T} \sum_{w \in V} p(w|x_{1:i-1}) \log p(w|x_{1:i-1}) \tag{19}$$

Let $\frac{1}{T} H_p[X] = \overline{H}_p[X]$ , i.e., the per word entropy. Let $X$ be the standard Penn Treebank test set, and $X_{\text{rev}}$ the reversed one. For a given probabilistic model $p$, we define the entropy gap $\Delta H_p$ by

$$\Delta H_p = \overline{H}_p[X_{\text{rev}}] - \overline{H}_p[X]. \tag{20}$$

Since $X_{\text{rev}}$ clearly does not come from the training data distribution (reversed English does not look like proper English), we expect $\Delta H_p$ to be positive and large. Namely, if we take the per word entropy of a model as a proxy for the models' certainty (low entropy means the model is confident about its prediction), then the overall certainty of well calibrated models over $X_{\text{rev}}$ should be lower than over $X$. Thus, $\overline{H}_p[X_{\text{rev}}] > \overline{H}_p[X]$. When comparing two distributions, we expect the better calibrated one to have a larger $\Delta H_p$.

In Figure 3, we plotted $\Delta H_p$ for the BBB and the baseline dropout LSTM model. The BBB model has a gap of about 0.67 nats/word when taking 10 samples, and slightly below 0.65 when using the posterior mean. In contrast, the model using MC Dropout (Gal & Ghahramani, 2015) is less well calibrated and is below 0.58 nats/word. However, when "turning off" dropout (i.e., using the mean field approximation), $\Delta H_p$ improves to below 0.62 nats/word.

We note that with the empirical likelihood of the words in the test set with size $T$ (where for each word $w \in V$, $p(w) = \frac{(\# \text{ occurrences of } w)}{T}$), we get an entropy of 6.33 nats/word. The BBB mean model has entropy of 4.48 nats/word on the reversed set which is still far below the entropy we get by using the empirical likelihood distribution.

### 6.2 IMAGE CAPTION GENERATION

We also applied Bayes by Backprop for RNNs to image captioning. Our experiments were based upon the model described in Vinyals et al. (2016), where a state-of-the-art pre-trained convolutional

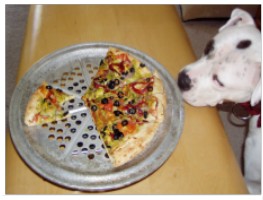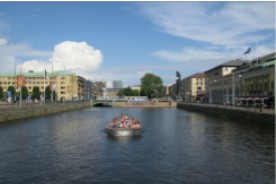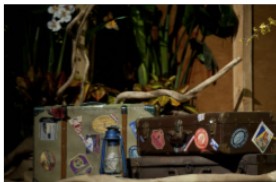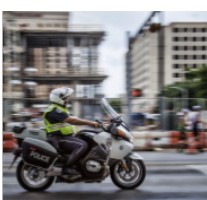

**Baseline:** a white plate with a pizza on it
**BBB:** a small white dog eating a piece of pizza

**Baseline:** a small boat in a large body of water
**BBB:** a boat traveling down a river next to a bridge

**Baseline:** a pile of luggage sitting on top of a wooden floor
**BBB:** a close up of a person holding a skate board

**Baseline:** a man riding a motorcycle down a street
**BBB:** a police officer riding a motorcycle down a street

Figure 4: Image captioning results on MSCOCO development set.

neural network (CNN) was used to map an image to a high dimensional space, and this representation was taken to be the initial state of an LSTM. The LSTM model was trained to predict the next word on a sentence conditioned on the image representation and all the previous words in the image caption. We kept the CNN architecture unchanged, and used an LSTM trained using Bayes by Backprop rather than the traditional LSTM with dropout regularisation. As in the case for language modelling, this work conveniently provides an open source implementation[2]. We used the same prior distribution on the weights of the network (18) as we did for the language modelling task, and searched over the same hyper-parameters.

We used the MSCOCO (Lin et al., 2014) data set and report perplexity, BLUE-4, and CIDER scores on compared to the Show and Tell model (Vinyals et al., 2016), which was the winning entry of the captioning challenge in 2015[3]. The results are:

| Model | Perplexity | BLUE-4 | CIDER |
|---|---|---|---|
| Show and Tell | 8.3 | 28.8 | 89.8 |
| Bayes RNN | **8.1** | **30.2** | **96.0** |

We observe significant improvements in BLUE and CIDER, outperforming the dropout baseline by a large margin. Moreover, a random sample of the captions that were different for both the baseline and BBB is shown in Figure 4. Besides the clear quantitative improvement, it is useful to visualise qualitatively the performance of BBB, which indeed generally outperforms the strong baseline, winning in most cases. As in the case of Penn Treebank, we chose a performant, open source model. Captioning models that use spatial attention, combined with losses that optimise CIDER directly (rather than a surrogate loss as we do) achieve over 100 CIDER points (Lu et al., 2016; Liu et al., 2016).

## 7 DISCUSSION

We have shown how to apply the Bayes by Backprop (BBB) technique to RNNs. We enhanced it further by introducing the idea of posterior sharpening: a hierarchical posterior on the weights of neural networks that allows a network to adapt locally to batches of data by a gradient of the model.

We showed improvements over two open source, widely available models in the language modelling and image captioning domains. We demonstrated that not only do BBB RNNs often have superior performance to their corresponding baseline model, but are also better regularised and have superior uncertainty properties in terms of uncertainty on out-of-distribution data. Furthermore, BBB RNNs through their uncertainty estimates show signs of knowing what they know, and when they do not, a critical property for many real world applications such as self-driving cars, healthcare, game playing, and robotics. Everything from our work can be applied on top of other enhancements to RNN/LSTM models (and other non-recurrent architectures), and the empirical evidence combined with improvements such as posterior sharpening makes variational Bayes methods look very promising. We are exploring further research directions and wider adoption of the techniques presented in our work.

---

[2] https://github.com/tensorflow/models/tree/master/im2txt
[3] The winning entry was an ensemble of many models, including some with fine tuning w.r.t. the image model. In this paper, though, we report single model performance.

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

# A  SUPPLEMENTARY MATERIAL

## A.1  BAYES BY BACKPROP ALGORITHM

Algorithm 2 shows the Bayes by Backprop Monte Carlo procedure from Section 2 for minimising the variational free energy from Eq. (1) with respect to the mean and standard deviation parameters of the posterior $q(\theta)$.

---

**Algorithm 2** Bayes by Backprop

---

Sample $\epsilon \sim \mathcal{N}(0, I)$, $\epsilon \in \mathbb{R}^d$.
Set network parameters to $\theta = \mu + \sigma\epsilon$.
Do forward propagation and backpropagation as normal.
Let $g$ be the gradient w.r.t. $\theta$ from backpropagation.
Let $g_\theta^{KL}, g_\mu^{KL}, g_\sigma^{KL}$ be the gradients of $\log \mathcal{N}(\theta|\mu, \sigma^2) - \log p(\theta)$ with respect to $\theta$, $\mu$ and $\sigma$ respectively.
Update $\mu$ according to the gradient $g + g_\theta^{KL} + g_\mu^{KL}$.
Update $\sigma$ according to the gradient $(g + g_\theta^{KL})\epsilon + g_\sigma^{KL}$.

---

## A.2  LSTM EQUATIONS

The core of an RNN, $f$, is a neural network that maps the RNN state at step $t$, $s_t$ and an input observation $x_t$ to a new RNN state $s_{t+1}$, $f : (s_t, x_t) \mapsto s_{t+1}$.

An LSTM core Hochreiter & Schmidhuber (1997) has a state $s_t = (c_t, h_t)$ where $c$ is an internal core state and $h$ is the exposed state. Intermediate gates modulate the effect of the inputs on the outputs, namely the input gate $i_t$, forget gate $f_t$ and output gate $o_t$. The relationship between the inputs, outputs and internal gates of an LSTM cell (without peephole connections) are as follows:

$$
\begin{aligned}
i_t &= \sigma(W_i[x_t, h_{t-1}]^T + b_i), \\
f_t &= \sigma(W_f[x_t, h_{t-1}]^T + b_f), \\
c_t &= f_t c_{t-1} + i_t \tanh(W_c[x_t, h_{t-1}] + b_c), \\
o_t &= \sigma(W_o[x_t, h_{t-1}]^T + b_o), \\
h_t &= o_t \tanh(c_t),
\end{aligned}
$$

where $W_i$ $(b_i)$, $W_f$ $(b_f)$, $W_c$ $(b_c)$ and $W_o$ $(b_o)$ are the weights (biases) affecting the input gate, forget gate, cell update, and output gate respectively.

## A.3  WEIGHT PRUNING

As discussed in Section 6.1, for the Penn Treebank task, we have taken the converged model an performed weight pruning on the parameters of the network. Weights were ordered by their signal-to-noise ratio ($|\mu_i|/\sigma_i$) and removed (set to zero) in reverse order. It was observed that around 80% of the weights can be removed from the network with little impact on validation perplexity. In Figure 5, we show the patterns of the weights dropped for one of the LSTM cells from the model.

## A.4  ADDITIONAL CAPTIONING EXAMPLES

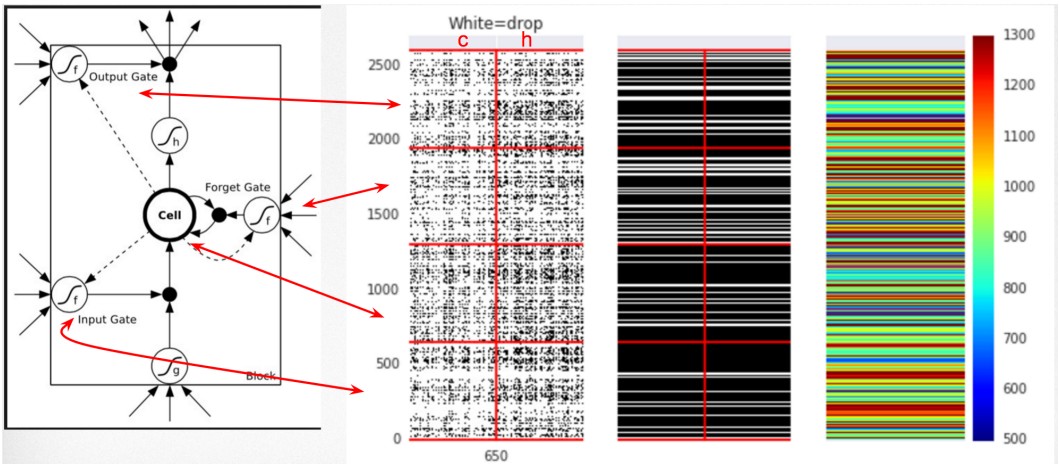

Figure 5: Pruning patterns for one LSTM cell (with 650 untis) from converged model with 80% of total weights dropped. A white dot indicates that particular parameter was dropped. In the middle column, a horizontal white line means that row was set to zero. Finally, the last column indicates the total number of weights removed for each row.

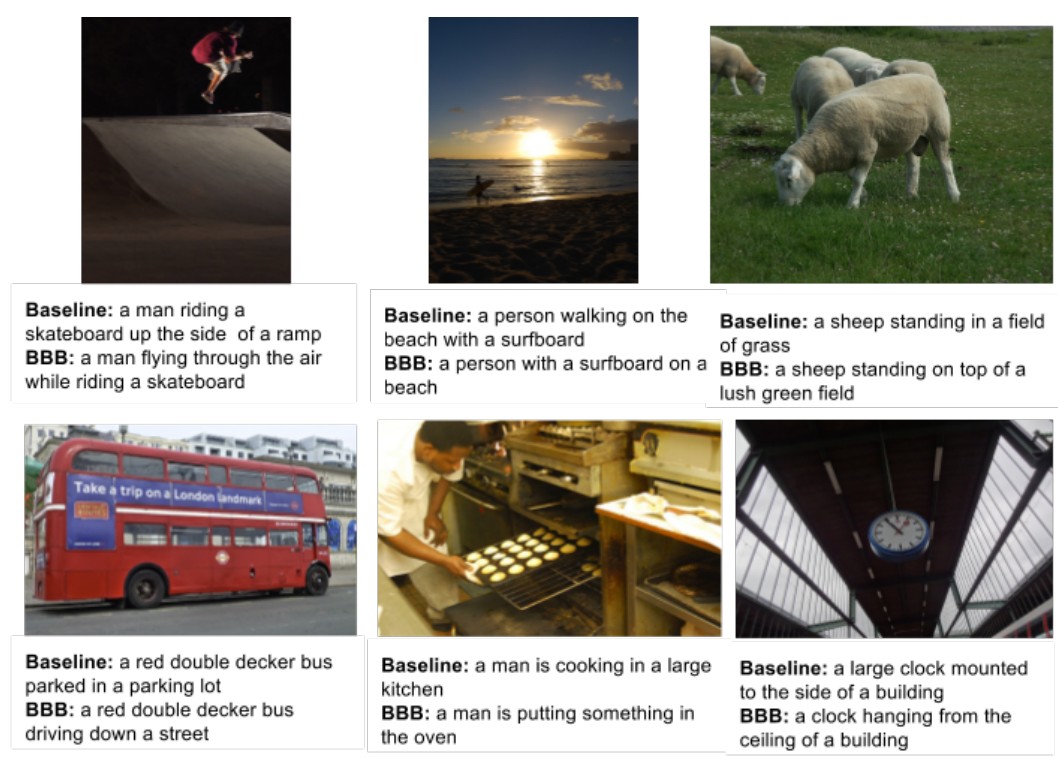

Figure 6: Additional Captions from MS COCO validation set.

