# OpenReview forum: "Revisiting Bayes by Backprop"
_ICLR.cc/2018/Conference — Reject_

### Official Review · AnonReviewer3 · 2017-11-27
**Variational inference + reparameterisation trick for Bayesian recurrent neural networks**

**Rating:** 5
**Confidence:** 4

**Review:**

*Summary*

The paper applies variational inference (VI) with the 'reparameterisation' trick for Bayesian recurrent neural networks (BRNNs). The paper first considers the "Bayes by Backprop" approach of Blundell et al. (2015) and then modifies the BRNN model with a hierarchical prior over the network parameters, which then requires a hierarchical variational approximation with a simple linear recognition model. Several experiments demonstrate the quality of the prediction and the uncertainty over dropout.

*Originality + significance*

To my knowledge, there is no other previous work on VI with the reparameterisation trick for BRNNs. However, one could say that this paper is, on careful examination, an application of reparameterisation gradient VI for a specific application.

Nevertheless, the parameterisation of the conditional variational distribution q(\theta | \phi, (x, y)) using recognition model is interesting and could be useful in other models. However, this has not been tested or concretely shown in this paper. The idea of modifying the model by introducing variables to obtain a looser bound which can accommodate a richer variational family is also not new, see: hierarchical variational model (Ranganath et al., 2016) for example.

*Clarity*

The paper is, in general, well-written. However, the presentation in 4 is hard to follow. I would prefer if appendix A3 was moved up front -- in this case, it would make it clear that the model is modified to contain \phi, a variational approximation over both \theta and \phi is needed, and a q that couples \theta, \phi and and the gradient of the log likelihood term wrt \phi is chosen.

Additional comments:

Why is the variational approximation called "sharpened"?

At test time, normal VI just uses the fixed q(\theta) after training. It's not clear to me how prediction is done when using 'posterior sharpening' -- how is q(\theta | \phi, x) in eqs. 19-20 parameterised? The first paragraph of page 5 uses q(\theta | \phi, (x, y)), but y is not known at test time.

What is C in eq. 9?

This comment "variational typically underestimate the uncertainty in the posterior...whereas expectation propagation methods are mode averaging and so tend to overestimate uncertainty..." is not precise. EP can do mode averaging as well as mode seeking, depending on the underlying and approximate factor graphs. In the Bayesian neural network setting when the likelihood is factorised point-wise and there is one factor for each likelihood, EP is just as mode-seeking as variational. On the other hand, variational methods can avoid modes too, see the mixture of Gaussians example in the "Two problems with variational EM... " paper by Turner and Sahani (2010).

There are also many hyperparameters that need to be chosen -- what would happen if these are optimised using the free-energy? Was there any KL reweighting scheduling as done in the original BBB paper?

What is the significance of the difference between BBB and BBB with sharpening in the language modelling task? Was sharpening used in the image caption generation task?

What is the computational complexity of BBB with posterior sharpening? Twice that BBB? If this is the case, would BBB get to the same performance if we optimise it for longer? Would be interesting to see the time/accuracy frontier.

---

> ### Author Response · Authors · 2018-01-05
> **manuscript updated, further clarifications and comments on naming and computational cost**
>
> Thanks for helpful comments and useful feedback; we have made amendments the manuscript.
>
> We accepted the suggestion of moving Appendix A3 to the main text of paper, we agree that it makes the presentation more clear.
>
> Regarding the constant C, it is the number of truncated sequences, it is specified just above eq (4) in the paper. We have made it more explicit on the revised version.
>
> We thank the reviewer for the comment on mode seeking and move averaging, and have updated the text to be more precise.
>
> Regarding the choice of hyperparameters by using the free energy, we optimised the hyperparameters using the performance on the tasks we considered (perplexity); but we found this to correlate with the free-energy. Moreover, we did not do any KL reweighting scheduling.
>
> In terms of evaluation, many applications of language modeling (such as machine translation, or speech recognition) use a language model to “rank” sentences. In this case, “y” is known at test time. Otherwise, one can still use the hierarchical prior that does not depend on knowing the answer (to e.g. do ancestral sampling).
>
> The posterior sharpening technique was not tested in the image captioning task and still needs to be further investigated. The improvements of using the posterior sharpening technique are small (but consistent) when compared to standard BBB. Perhaps also shifting the variance of the posterior rather than only the mean (or instead of stepping in the direction of the gradient, you do an update RMS style as proposed in "Dynamic Evaluation of neural sequence models"  Krause et al) would yield further improvements.
>
> We may consider renaming posterior sharpening to posterior shifting as that more accurately describes the technique that we introduced in this paper. Furthermore, we believe the technique can still be enhanced by e.g. shifting the variance of the posterior rather than only the mean (or instead of stepping in the direction of the gradient, you do an update RMS style as proposed in "Dynamic Evaluation of neural sequence models"  Krause et al). Nonetheless, the small (but consistent) improvements shown in the paper and the VAE treatment of Bayesiann Neural Networks novel to this technique makes us excited for further developments around posterior sharpening / shifting.
>
>
> Regarding the computational cost for BBB with posterior sharpening, it will be twice as for standard BBB because the computational cost is dominated by the backward pass of the neural network and posterior sharpening requires two backward passes (see reply to AnonReviewer1 for further discussion). All reported performances are at convergence where both methods have remained at the same performance for the same amount of time. We observed it took roughly the same number of steps to plateau.

---

### Official Review · AnonReviewer1 · 2017-11-27
**Interesting posterior sharpening idea**

**Rating:** 6
**Confidence:** 4

**Review:**

This paper proposes an interesting variational posterior approximation for the weights of an RNN. The paper also proposes a scheme for assessing the uncertainty of the predictions of an RNN.

pros:
--I liked the posterior sharpening idea. It was well motivated from a computational cost perspective hence the use of a hierarchical prior.
--I liked the uncertainty analysis. There are many works on Bayesian neural networks but they never present an analysis of the uncertainty introduced in the weights. These works can benefit from the uncertainty analysis scheme introduced in this paper.
--The experiments were well carried through.

cons:
--Change the title! the title is too vague. "Bayesian recurrent neural networks" already exist and is rather vague for what is being described in this paper.
--There were a lot of unanswered questions:
 (1) how does sharpening lead to lower variance? This was a claim in the paper and there was no theoretical justification or an empirical comparison of the gradient variance in the experiment section
(2) how is the level of uncertainty related to performance? It would have been insightful to see effect of \sigma_0 on the performance rather than report the best result.
(3) what was the actual computational cost for the BBB RNN and the baselines?
--There were very minor typos and some unclear connotations. For example there is no such thing as a "variational Bayes model".

I am willing to adjust my rating when the questions and remarks above get addressed.

---

> ### Public Comment · (anonymous) · 2017-12-25
> **Uncertainty analysis has been been conducted**
>
> For the comments "There are many works on Bayesian neural networks but they never present an analysis of the uncertainty introduced in the weights."
>
> I am not sure whether this is true. [*] conducted  the uncertainty analysis in the context of RNNs. Please see details below:
>
> (1) In Figure 4: Image captioning with different weight samples (each sample is a RNN). It shows the diversity of generated captions due to the uncertainty in the weights. Left are the given images, right are the corresponding captions. The captions in each box are from the same model sample
>
> (2) In Figure 6: Question type classification. Both the mean and standard derivation of prediction are shown. It suggests one can leverage the uncertainty information to make decisions: either manually make a human judgement when uncertainty is high, or automatically choose the one with lower standard derivations when both types exhibits similar prediction means.
>
> [*] Scalable Bayesian Learning of Recurrent Neural Networks for Language Modeling, ACL 2017

---

> ### Author Response · Authors · 2018-01-05
> **paper title changed, some comments on variances and computational costs**
>
> Thanks for helpful comments and useful feedback; we have made amendments the manuscript.
>
> We agree that the title of the paper is too vague and have updated it to "Revisiting Bayes by Backprop".
>
> Regarding the lower variance of posterior sharpening, we point the reviewer to the discussion on the last paragraph of Session 6.1. There we compare the perplexity of standard training (i.e., deterministic weights), standard BBB approach and BBB with posterior shaperning after only one epoch of training. We see the model with posterior sharpening trains faster and achieves significantly better performance after one epoch, significantly closing the gap with standard training (zero variance) (perplexities of 205 (zero variance), 227 (posterior sharpening) vs 258 (standard BBB)).
>
> Regarding the effect of sigma_0 on the performance of posterior sharpening. We did not find sigma_0 to have a significant effect on performance: if sigma_0 is set too small (<10^-10), you recover the BBB baseline as the KL term pushes \eta towards 0; if sigma_0 is too large (>0.2), the noise in parameter space becomes too large and no training occurs. The effect is otherwise small but consistently outperforms the BBB baseline.
>
> Regarding the computational cost (at training time), as we stated towards the end of section 6.1: "
> we note that the speed of our naive implementation of Bayesian RNNs was 0.7 times the original
> speed and 0.4 times the original speed for posterior sharpening". Note that the asymptotic time complexity remains unchanged because the run time complexity of a forward and backward pass through the network is still dominated by the same computations as in a non-Bayesian RNN.

---

### Official Review · AnonReviewer2 · 2017-11-29
**Interesting change in inference but not clear why the modification helps**

**Rating:** 6
**Confidence:** 5

**Review:**

The manuscript proposes a new framework for inference in RNN based upon the Bayes by Backprop (BBB) algorithm.  In particular, the authors propose a new framework to "sharpen" the posterior.

In particular, the hierarchical prior in (6) and (7) frame an interesting modification to directly learning a multivariate normal variational approximation.  In the experimental results, it seems clear that this approach is beneficial, but it's not clear as to why.  In particular, how does the variational posterior change as a result of the hierarchical prior?  It seems that (7) would push the center of the variational structure back towards the MAP point and reduces the variance of the output of the hierarchical prior; however, with the two layers in the prior it's unclear what actually is happening.  Carefully explaining *what* the authors believe is happening and exploring how it changes the variational approximation in a classic modeling framework would be beneficial to understanding the proposed change and evaluating it.  As a final point, the authors state, "as long as the improvement along the gradient is great than the KL loss incurred...this method is guaranteed to make progress towards optimizing L."  Do the authors mean that the negative log-likelihood will be improved in this case?  Or the actual optimization?  Improving the negative log-likelihood seems straightforward, but I am confused by what the authors mean by optimization.

The new evaluation metric proposed in Section 6.1.1 is confusing, and I do not understand what the metric is trying to capture.  This needs significantly more detail and explanation.  Also, it is unclear to me what would happen when you input data examples that are opposite to the original input sequence; in particular, for many neural networks the predictions are unstable outside of the input domain and inputting infeasible data leads to unusable outputs.  It's completely feasible that these outputs would just be highly uncertain, and I'm not sure how you can ascribe meaning to them.  The authors should not compare to the uniform prior as a baseline for entropy.  It's much more revealing to compare it to the empirical likelihoods of the words.

---

> ### Author Response · Authors · 2018-01-05
> **manuscript updated and some clarifications.**
>
> Thanks for helpful comments and useful feedback; we have made amendments the manuscript.
>
> Regarding the posterior sharpening technique, we note that (7) pushes the mean of the posterior towards the maximum likelihood solution, not the MAP solution. Pushing towards the MAP solution is also an option, but as the reviewer notes, in the case of a hierarchical prior, a chicken-and-egg problem emerges as the posterior is defined in terms of the posterior sharpening already. The classic variational formulation for posterior sharpening was previously in Appendix A3 and A4, but now it has been moved to the main text (Sec 4.1-4.2) as suggested by AnonReviewer3.
>
> Regarding the statement "as long as the improvement along the gradient is great than the KL loss incurred...this method is guaranteed to make progress towards optimizing L." Thanks for pointing out the lack of clarity. What we meant is: if the gradient g_phi improves the log likelihood log p(y|theta,x) term more than the KL cost added for posterior sharpening (KL[q(theta|phi,(x,y))||p(theta|phi)]) then the lower bound in (8) will improve. We have amended it in the manuscript.
>
> Regarding the evaluation metric in 6.1.1, the intuition behind it is if you take a natural language sentence and reverse it, then this destroys much of its structure. One would expect that a probabilistic language model (LM) would give lower probability to the reversed sentence over the original. Moreover, a LM equipped with uncertainty estimates such as the one proposed here should produce lower certainty for out of domain inputs (such as reversed text). The metric precisely tries to quantify this (un)certainty. This was meant to be a very simple illustration of how uncertainty estimates behave when the language models are misspecified. Finally, we agree that comparing to the empirical likelihoods is more sensible and we have updated the manuscript with it.

---

### Public Comment · (anonymous) · 2017-11-27
**Is it possible to revise the title?**

Is it possible to revise the title, to better reflect the proposed variational technique for RNNs? "Bayesian Recurrent Neural Networks" have been proposed in several papers with different Bayesian learning methods. See below for examples:

A Theoretically Grounded Application of Dropout in Recurrent Neural Networks, NIPS 2017
Scalable Bayesian Learning of Recurrent Neural Networks for Language Modeling, ACL 2017
Bayesian Recurrent Neural Network for Language Modeling, IEEE Trans Neural Netw Learn Syst. 2016

---

### Decision · Program_Chairs · 2018-01-29
**ICLR 2018 Conference Acceptance Decision**

**Decision:**

Reject

**Comment:**

Thank you for submitting you paper to ICLR. The revision improved the paper e.g. moving Appendix A3 to the main text has improved clarity, but, like reviewer 3, I still found section 4 hard to follow. As the authors suggest, shifting the terminology to "posterior shifting” rather than “sharpening" would help at a high level, but the design choices should be more carefully explained. The experiments are interesting and promising. The title, although altered, still seems a misnomer given that the experimental evaluation focusses on RNNs.

Summary: There is the basis of a good paper here, but the rationale for the design choices should be more carefully explained.